# meal&me : Leveraging Semantic Technologies for Personalized Restaurant Recommendation

Davide Buscaldi[1], Hamid Hammouche[2], and Didier Cholvy[2]

LIPN, Université Sorbonne Paris Nord, Villetaneuse, France
davide.buscaldi@lipn.univ-paris13.fr
Advanced Decision, 243 Bis, Boulevard Pereire, Paris, France
{hha,dch}@advanceddecision.fr

## 1   Introduction

meal&me is a restaurant recommendation service based on Natural Language Processing (NLP) guided analysis of lived experiences in publicly accessible reviews. The innovation of meal&me consists in the fact that the user experiences are analyzed at a fine-grained level, taking into account real experiences in restaurants and matched to users' implicit wishes. The integration of a semantic understanding of customer experiences is, in our opinion, the key to making highly personalized recommendations. To implement this semantic understanding we applied an innovative method for aspect-based sentiment analysis [2] and knowledge engineering. One of the key design issues consists in retrieving targeted aspects from reviews. The aspects are defined combining our ontology of *experiences* with the DATAtourisme Ontology [1], issued from the DATA-tourisme project, a national French project whose aim is to create an Open Data platform for the collection and distribution of data related to the points of interest of touristic industry (type of cuisine, offers, opening hours). In our proprietary ontology we defined 15 aspects to model characteristics for restaurants such as ambiance, surroundings, and service. Each of these aspects is evaluated in a range of 5 scores from very negative to very positive by the NLP-guided fine-grained sentiment analysis system. One problem in evaluating the reviews is that some of the aspects are indirectly addressed by evaluating not the aspect directly, but part of it: for instance, the aspect *ambiance* can be addressed by evaluating the decoration of the restaurant, the vibe and so on. Therefore, we had to carry out an ontology enrichment process, guided by the content of reviews, which allowed us to define 10 sub-aspects for each of the 15 aspects that refine the knowledge about them.

## 2   Enrichment of the DATAtourisme Ontology

DATATourisme is an Ontology written in OWL that contains information about the touristic point of interests in different regions of France. For our work, we were interested only to the part of the ontology related to restaurants. The information provided includes the geographical coordinates, contact numbers,

capacity, and more. However, since the ontology is the result of the aggregation of 45 different data sources, it has some problems with homogeneity: different regions provide different information arranged in different ways. It is also a middle-level ontology that is not particularly fit to a specific task. Therefore, we had to improve DATAtourisme to be able to use it in our service, and e improved it in two ways: first, we refined the *aspect* sub-ontology by introducing new aspects. Then, we created a new concept, the user *experience* as a composition of the restaurant aspects. For instance, the experience *business meal* would require a classy ambiance, excellent service, stylish decoration and creative cuisine.

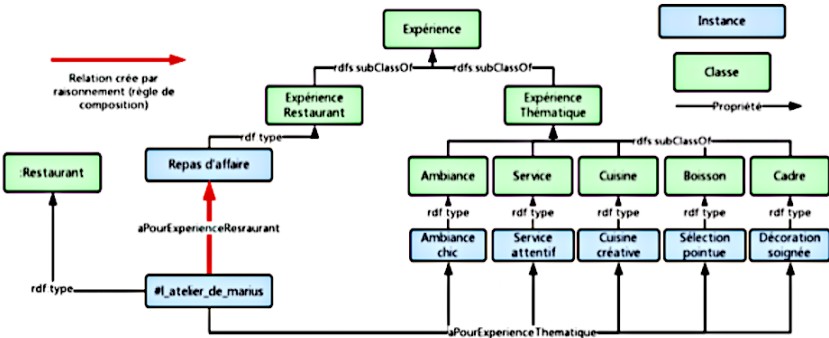

**Fig. 1.** An exmple of Experience modelled from the enriched DATAtourisme ontology. In red, the relation built by inference.

From a manually annotated corpus of $8,381$ reviews in French of Parisian restaurants, we extracted the most frequent n-grams for each of the core aspects. For example, in the case of the aspect *ambiance*, we obtained "Super soirée" (super night), "ambiance conviviale", "ambiance decontractée", "ambiance détendue" (pleasant, relaxed ambient), among others. The creation of the experiences themselves was done manually as a set of inference rules, written as composition rules using the the Apache Jena[1] inference API.

## 3   Conclusion

The use of semantic technologies has been key in the implementation of meal&me and we could say that the application could not exist without them. As future works, we would like to automate the ontology enrichment and population processes, and to use Neural Networks methods that should be able to mitigate the problems that usually affect sentiment analysis, such as the presence of negations and irony.

---

[1] https://jena.apache.org/

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

## Annex

Our application is currently under deployment in the Paris region. We show in this annex some screen captures of the application features.

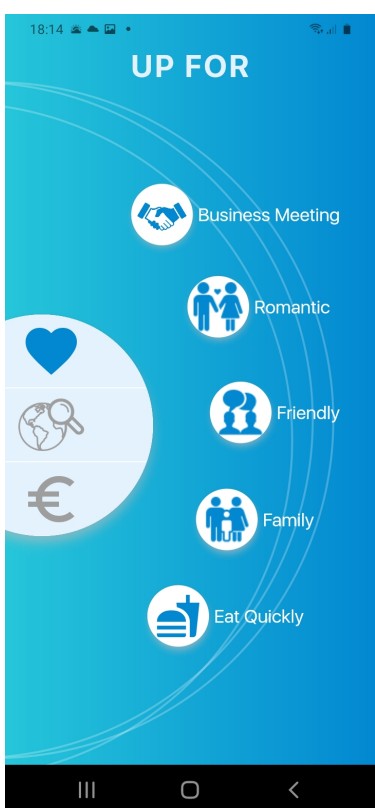
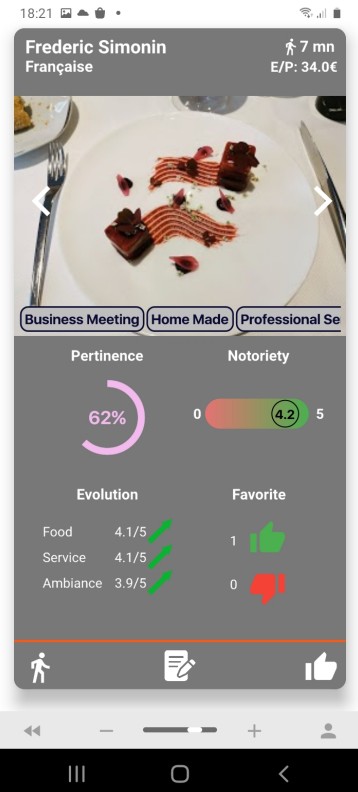

**Fig. 2.** Left: Welcome screen with a choice of experiences. Right: Example of restaurant entry with aspect scores, pertinence and notoriety.