# OpenReview forum: "meal&me : Leveraging Semantic Technologies for Personalized Restaurant Recommendation"
_eswc-conferences.org/ESWC/2021/Conference/Industry_Track — Submitted to ESWC 2021 Industry_

### Official Review · ~Kimberly_Garcia1 · 2021-04-13
**meal&me Leveraging Semantic Technologies for Personalized Restaurant Recommendations**

**Rating:** 7
**Confidence:** 4

**Review:**

This extended abstract presents a propiertary ontology for describing restaurants according to the experience they offer. Such an ontology is based on the DATAtourisme ontology, which is a French initiative for tourism linked data.  Such a foundation ontology was extended with 15 classes derived from an NLP sentiment analysis of reviews of restaurant located in the Parisian region.

This work is interesting, since it shows an approach to automatically enrich manually crafted ontologies with numeric approaches (such as NLP). The work has a nice structure that it is easy to read and follow. However, some clarifications in the implementation could make the text stronger: a) how did you decide to add another “Experience” subclass? How many occurrences were found in the restaurant reviews/What is the criteria to consider the  “Cuisine Creative” and “Service attentif” as absolutely necessary subclasses?; b) at the end of section 2, it is not clear what the manual part that was created using inference rules is. Was it the creation of the actual experience subclasses? Or the placing of a specific restaurant instance in one of those subclasses; c) how scalable is your approach? When a new restaurant opens how long will it take until it can be categorised and shown in the app?; d) have you done any tests to validate your results i.e., are restaurant accurately described given the reviews analysis?

As DATAtourisme states in their website the reuse of ontologies, schemas and vocabularies is one of the main practices of linked data. Have you consider making available your ontology for other developers/knowledge engineers to reuse?

---

### Official Review · ~Yushan_Liu1 · 2021-04-15
**Promising application with need for more detailed documentation**

**Rating:** 5
**Confidence:** 4

**Review:**

The authors introduce a restaurant recommendation application called meal&me that makes use of semantic information about user experiences and wishes for more personalized recommendation.
They extend an existing tourism ontology, DATAtourisme Ontology, to include several aspects describing user experiences.

Strengths:
- Interesting and relevant topic of personalized restaurant recommendation that takes real user experiences in the form of reviews (using NLP and sentiment analysis) into account.
- Fine-grained description of restaurants (15 aspects with 10 sub-aspects each) in an extended ontology.
- First deployment in the Paris region.

Weaknesses:
- It is not well documented how exactly the app should be used or which detailed features the app provides.
- There is no evaluation of the deployment or impact of the app, e.g., user satisfaction, number of downloads.
- Fig. 1 is slightly blurred.
- Several typos and grammatical errors in the text.

Due to the lack of a more detailed documentation of the app as well as the (potential) impact after deployment, I recommend a weak rejection.

---

### Official Review · ~Anna_Himmelhuber1 · 2021-04-16
**Interesting idea, some room for improvement**

**Rating:** 5
**Confidence:** 3

**Review:**


The paper describes an approach integrating  semantic understanding of customer experiences is for personalized recommendations, which is an interesting idea.

The implementation is done by aspect-based sentiment analysis and knowledge engineering, and some details are given regarding the enrichment process of the ontology.

However, how the restaurant reviews will be matched to users’ implicit wishes isn't explained. As this framework is currently under deployment, some at least preliminary results including user acceptability would have been nice included in order to assess the method.  Additionaly, there are some possibilities for improvement regarding the quality of the figure, which is hard to read as well as the language in terms of grammar and choice of words (e.g. notoriety)

Overall, the paper is a bit premature as essential parts of the process and first results are missing.